# Relationship between Anxiety, Depression and Learning Burnout of Nursing Undergraduates after the COVID-19 Epidemic: The Mediating Role of Academic Self-Efficacy

**DOI:** 10.3390/ijerph20054194

**Published:** 2023-02-26

**Authors:** Pingting Zhu, Ting Xu, Huiwen Xu, Qiaoying Ji, Wen Wang, Meiyan Qian, Guanghui Shi

**Affiliations:** School of Nursing and School of Public Health, Yangzhou University, Yangzhou 225009, China

**Keywords:** COVID-19 epidemic, normalization, nursing undergraduate, academic self-efficacy, learning burnout, anxiety, depression

## Abstract

Background: Although cross-sectional studies on the learning status of nursing undergraduates during the COVID-19 epidemic have surged, few studies have explored the normalization of COVID-19 on students’ learning burnout and mental health. The study was designed to investigate the learning burnout of nursing undergraduates in school under the normalization of the COVID-19 epidemic and explore the hypothesized mediation effect of academic self-efficacy in the relationship between anxiety, depression and learning burnout in Chinese nursing undergraduates. Methods: A cross-sectional study was conducted among nursing undergraduates in the school of nursing of a university in Jiangsu Province, China (*n* = 227). A general information questionnaire, College Students’ Learning Burnout Questionnaire, Generalized Anxiety Disorder Scale (GAD-7), and Patient Health Questionnaire depression scale (PHQ-9) were administered. Descriptive statistical analysis, Pearson correlation analysis, and multiple linear regression analysis were performed via SPSS 26.0. Process plug-in (Model 4) was used to test the mediating effect of academic self-efficacy (bootstrap 5000 iterations, α = 0.05). Results: Learning burnout (54.1 ± 0.656) was positively correlated with anxiety (4.6 ± 0.283) and depression (5.3 ± 0.366) (*p* < 0.01) and was negatively correlated with academic self-efficacy (74.41 ± 0.674) (*p* < 0.01). Academic self-efficacy plays a mediating role between anxiety and learning burnout (0.395/0.493, 80.12%) and a mediating role between depression and learning burnout (0.332/0.503, 66.00%). Conclusion: Academic self-efficacy has a significant predictive effect on learning burnout. Schools and teachers should strengthen the screening and counselling of students’ psychological problems, detect learning burnout caused by emotional problems in advance and improve students’ initiative and enthusiasm for learning.

## 1. Introduction

The impact of the novel coronavirus pneumonia epidemic (COVID-19) on the world’s education system continues to spread, especially for medical education [1,2]. During the widespread outbreak of the COVID-19 pandemic, medical clinical practice was forced to be terminated to protect the lives, health, and safety of medical students, and replaced with online teaching [3,4,5]. Online teaching has become the preferred teaching method to deal with the normalization of the COVID-19 epidemic. However, although online teaching has advantages, it still cannot completely replace traditional offline teaching [6,7,8,9]. The teaching effect is affected by a variety of subjective and objective factors, which are first reflected in the subjective responses of students. Studies have shown that COVID-19 affects students’ learning behavior and mental health, leading to learning burnout and reduced learning efficiency, affecting academic performance [10,11,12]. The education of medical students is particularly important for their future career development and tasks [13,14]. The emergence and aggravation of learning burnout have aroused the wide attention of medical education scholars [15,16].

Learning burnout is a psychological distress problem, which is defined as low mood and low learning efficiency caused by greater academic pressure and unrealized academic requirements [17]. Learning burnout not only affects students’ academic performance and directly leads to a serious decline in academic quality, but also affects students’ physical and mental health and future career development [18]. Higher levels of learning burnout can even lead to underachievement or dropout, affecting nursing students’ graduation and employment. Nursing and clinical medicine are inseparable and the associated academic pressure is higher than that of other majors [19,20]. The learning psychological state of nursing undergraduates has aroused widespread concern in the education sector.

The mental health of nursing students is the focus of nursing education research, especially its impact on theoretical learning and clinical practice. Nursing students are mainly affected by two kinds of pressure. The first is academic pressure, including theoretical courses and performance assessments, such as mid-term assessments, professional qualification examinations, graduation tests, recruitment examinations and entrance examinations [21,22]. The second is the pressure of clinical practice, mainly including the lack of clinical ability, the gap between theory and practice, the care of patients, handling medical devices, the assessment of interns, interpersonal communication, the criticism of teachers and the psychological stress caused by rescuing patients [23,24,25,26]. All kinds of psychological problems caused by academic studies and practice are more prominent among nursing students, and these psychological problems are important factors leading to career change in nursing students and the resignation of newly graduated nurses [27,28,29,30].

At the beginning of the COVID-19 outbreak, the anxiety (55.9%) and depression (52%) symptoms of nursing students seriously affected their learning psychological state and academic performance, and some unresolved anxiety and depression problems had an impact on their learning until the late epidemic [31,32]. Affected by COVID-19, the learning burnout of college students in Asia is the most serious. The fear, anxiety, and depression caused by COVID-19, the burnout caused by long-term use of electronic equipment, unsuitable learning environments and the problems related to poor Internet connection when participating in online teaching directly or indirectly affect the learning situation of students, increasing the level of learning burnout and affecting academic performance [6,9,32,33,34,35]. The learning burnout caused by COVID-19 has a particularly serious impact on nursing students. The lack of theoretical knowledge accumulation and clinical practice skills have left hidden dangers for their future competence in nursing positions [14,23]. Therefore, it is particularly important to detect learning burnout early and take effective intervention measures to promote the mental health of nursing students and improve their academic performance. Fortunately, students’ learning burnout may be affected by their academic self-efficacy. This finding provides an entry point for nursing educators to solve the problem of nursing students’ learning burnout [36,37].

Academic self-efficacy is the performance of self-efficacy in learning, which refers to the belief that the individual’s confidence or belief in the ability to achieve a specific goal will affect the individual’ s motivation and enthusiasm [38]. Bandura’s self-efficacy theory finds that students with low self-efficacy tend to compromise and give up when they encounter difficulties, which is the manifestation of learning burnout [38]. Belief in their ability to complete learning tasks will improve students’ self-efficacy and reduce the impact of learning burnout on achieving goals. However, academic self-efficacy will be affected by mental health as well as learning burnout [39,40,41,42]. Higher levels of distress will aggravate students’ psychological burden, affect their confidence in completing learning goals and tasks, and reduce their academic self-efficacy, which will lead to a series of learning psychological problems (such as learning burnout) and ultimately lead to a decline in academic performance [41,42,43].

During the normalization period of the COVID-19 epidemic, students’ academic self-efficacy has attracted much attention from educators. How to improve students’ academic self-efficacy has become a hot topic in the education field under the continuous influence of COVID-19. What is the mental health status of nursing students at present? How about their learning psychology and academic self-efficacy? Therefore, this study aims to investigate the mental health (anxiety and depression), learning burnout and academic self-efficacy of nursing students in Chinese universities under the normalization of COVID-19 and to explore the influence of academic self-efficacy on anxiety, depression and learning burnout of nursing undergraduates. In summary, we propose the following hypothesis:

**H1** : *Anxiety and depression should be directly related to learning burnout. The higher the anxiety and depression levels of nursing students, the more serious the learning burnout.*

**H2** : *Anxiety and depression should be directly related to academic self-efficacy. The higher the scores of anxiety and depression of nursing students, the lower their self-efficacy.*

**H3** : *The lower degree of anxiety and depression should be related to the lighter degree of learning burnout through stronger academic self-efficacy. The academic self-efficacy of nursing students plays an intermediary role between anxiety, depression and learning burnout.*

## 2. Methods

### 2.1. Study Design, Setting and Participants

Due to the normalized management of COVID-19, we conducted a cross-sectional study using an anonymous, self-made network questionnaire to reduce the risk of infection in face-to-face contact from April to May 2022. A cluster random sampling method was used to investigate nursing undergraduates from grade 1 to grade 4 in a school of nursing in Jiangsu Province, China (‘Questionnaire Star’, https://www.wjx.cn/newwjx/manage/myquestionnaires.aspx?randomt=1675663026 (accessed on 8 June 2022)). We conducted a pre-survey on 40 students (10 in each grade) randomly selected from freshmen to seniors and the final questionnaire was revised based on the data received from the pre-survey and the feedback from the students to ensure the comprehensibility and accuracy of each item of the questionnaire. Questionnaires were posted on WeChat before online teaching and participants filled out the questionnaires by using their mobile phones or computers. Before students filled in the questionnaire, they were informed of the purpose of this study and the principle of confidentiality of data. The home page of the questionnaire includes the voluntary nature of student participation and an informed consent statement. Students who completed the survey were considered to have agreed to participate in the study. All questionnaires were filled out voluntarily and submitted anonymously. The collected data are stored in the password-protected questionnaire star network questionnaire survey platform, which can only be accessed by the author. In addition, the students who participated in this study did not receive any rewards after completing the questionnaire.

Participants’ inclusion criteria: age > 18 years; full-time nursing undergraduates from freshman to senior. Exclusion criteria: non-nursing undergraduates; have a history of mental illness; participated in a similar survey in the past month; in the home or off-campus isolation phase.

### 2.2. Measurements

#### 2.2.1. General Information Questionnaire

The self-made general information questionnaire was used to investigate the gender, age, grade, place of origin, whether they were the only child, anyone in the medical profession in the family, monthly income per capita, academic performance and professional preference of nursing undergraduates.

#### 2.2.2. College Students’ Learning Burnout Scale

The revised scale was adopted based on the Maslach scale with three dimensions of low mood (eight items), improper behavior (six items) and low achievement (six items), a total of 20 items [44]. Using the Likert 5-level scoring method, the higher the score, the higher the degree of individual learning burnout. According to the total score to determine the degree of individual learning burnout: no learning burnout (≤40 points); difficult to determine (40–60 points); there is learning burnout (≥60 points). In this study, the Cronbach ‘s alpha of the scale is 0.87 and the Cronbach ‘s alpha of the three dimensions is 0.831 (low mood), 0.721 (improper behavior) and 0.70 (low achievement) respectively.

#### 2.2.3. Generalized Anxiety Disorder-7 (GAD-7)

The scale was compiled by Spitzer et al., including 7 items (0–3 each item), with a total score of 0–21 points [45]. The higher the score, the more severe the symptoms of individual anxiety. The score is divided into different levels of anxiety: no anxiety (0–4 points); may have mild anxiety (5–9 points); may have moderate anxiety (10–14 points); may have severe anxiety (≥15 points). The Cronbach’s alpha of the scale in this study was 0.91.

#### 2.2.4. Patient Health Questionnaire-9 (PHQ-9)

PHQ-9 was developed by Spitzer and contains 9 items (each score of 0–3 points), with a total score of 0–27 points [46]. The higher the score, the more serious the symptoms of individual depression. The score is divided into different levels of depression: no depression (0–4 points); may have mild depression (5–9 points); may have moderate depression (10–14 points); may have moderate to severe depression (15–19 points); may have severe depression (20–27 points). The Cronbach’s alpha of the scale in this study was 0.913.

#### 2.2.5. Academic Self-Efficacy Scale

The scale was compiled by Liang with 22 items [47]. It includes two dimensions of college students’ academic ability self-efficacy and academic behavior self-efficacy. Using the Likert 5-level scoring method, the higher the score, the higher the individual’ s learning self-efficacy, a method widely used in college students. In this study, the Cronbach’s alpha of the scale is 0.89 and the Cronbach’s alpha of the two dimensions is 0.912 (academic ability self-efficacy) and 0.789 (academic behavior self-efficacy).

### 2.3. Statistical Methods

SPSS 26.0 was used to analyze the data. Descriptive statistical analysis, Pearson correlation analysis and multiple linear regression analysis were performed. Based on controlling related factors, Model 4 in Process plug-in was used to analyze the mediating effect of academic self-efficacy on anxiety, depression and learning burnout. The Bootstrap method (Random 5000 times) was used to test the results and calculate the 95% confidence interval. When the confidence interval does not include 0, the statistical results are considered significant (α = 0.05).

### 2.4. Ethical Considerations

This study protocol is in line with the Helsinki Declaration and is authorized by the Ethics Committee of Yangzhou University College of Nursing (No. YZUHL20220029).

## 3. Results

### 3.1. General Characteristics

A total of 227 (89.7%) valid questionnaires were collected in this study and 26 invalid questionnaires were excluded. The general demographic data of specific nursing students are shown in Table 1. It can be seen that there are significant differences in learning burnout scores within different grades (*p* < 0.05), student’s academic performance *(p* < 0.01), preference for the nursing profession (*p* < 0.01) and whether the students’ relatives are engaged in medicine (*p* < 0.05). The scores for anxiety, depression, learning burnout, and academic self-efficacy of nursing undergraduates are shown in Table 2. Among them, no anxiety, mild, moderate and severe anxiety students accounted for 53.3%, 36.6%, 6.6% and 3.5%; the proportion of students without depression, mild, moderate, moderate and severe depression was 55.1%, 21.6%, 16.7%, 4.4% and 2.2%, respectively. The average score for learning burnout was 54.10 ± 0.656, with the highest score for low mood (21.07 ± 0.336). The average score of academic self-efficacy was 74.41 ± 0.674, and the score of learning ability self-efficacy was the highest (37.72 ± 0.419).

### 3.2. Correlation Analysis

Correlation analysis showed that learning burnout was positively correlated with anxiety and depression (r = 0.197, r = 0.275, *p* < 0.01) and negatively correlated with academic self-efficacy (r = −0.241, *p* < 0.01). Academic self-efficacy was negatively correlated with anxiety, depression and learning burnout (r = −0.240, r = −0.478, *p* < 0.01). There was a positive correlation between anxiety and depression (r = 0.778, *p* < 0.01). (Table 2)

### 3.3. Mediation Analysis

According to Harman’s single-factor test, there are 12 factors with eigenvalue > 1 in this study and the variation explained by the first factor is 24.29% (<40%), indicating that there is no common method deviation in the research results. Bootstrap was used to test the mediating role of academic self-efficacy in anxiety, depression and learning burnout. The results of linear regression analysis and mediating effect value and confidence interval between variables are shown in Table 3.

As can be seen from Figure 1, the effect of Anxiety on Academic self-efficacy was significant (a = −0.615, *p* < 0.001, 95%CI: −0.917, −0.313); the effect of Academic self-efficacy on Learning burnout was significant (b = −0.613, *p* < 0.001, 95%CI: −0.741, −0.545); the direct effect of Anxiety on Learning burnout was non-significant (c’ = 0.098, *p* > 0.05, 95%CI: −0.135, 0.331). Since the total effect of Anxiety on Learning burnout became significant after adding the mediator of Academic self-efficacy (c = 0.493, *p* < 0.01, 95%CI: 0.197, 0.790) and the indirect effect of Academic self-efficacy was significant (a * b = c-c’ = 0.395, *p* < 0.001) with 95% confidence interval excluding 0 (95%CI: 0.203, 0.606), academic self-efficacy played a fully mediating role in the association of anxiety with Learning burnout.

Figure 2 shows that the effect of Depression on Academic self-efficacy was significant (a = −0.53, *p* < 0.001, 95%CI: −0.761, −0.298); the effect of Academic self-efficacy on Learning burnout was significant (b = −0.627, *p* < 0.001, 95%CI: −0.725, −0.529); the direct effect of Depression on Learning burnout was non-significant (c’ = 0.171, *p* > 0.05, 95%CI: −0.010, 0.352). After adding Academic self-efficacy, the total effect of Depression on Learning burnout became significant (c = 0.503, *p* < 0.001, 95%CI: 0.277, 0.729) and the indirect effect of Academic self-efficacy was significant (a * b = c-c’ = 0.332, *p* < 0.001) with 95% confidence interval excluding 0 (95%CI: 0.178, 0.515), so academic self-efficacy played a fully mediating role in the association of depression with Learning burnout.

## 4. Discussion

In this study, the vast majority (74%) of nursing undergraduates acknowledged the existence of learning burnout under the normalization of COVID-19 and the degree of learning burnout was above average (54.10 ± 0.656). Only 7.92% of the students reported that there may be no learning burnout, 62.56% of the students found it difficult to determine whether there is learning burnout, and 29.52% of the students may have learning burnout. Compared with the learning burnout of nursing students during the COVID-19 pandemic (2.97 ± 0.34), the average score of learning burnout of nursing students in this study decreased (2.71 ± 0.033), which was also lower than that before the COVID-19 [15,37,48,49]. The difference in the results between the studies may be due to the low adaptability of students during the COVID-19 pandemic, including the psychological adaptability of public health emergencies, the adaptability of online learning and the poor learning environment [9,50,51]. Under the normalization of COVID-19, the form and content of online teaching are improving day by day and students’ adaptability and satisfaction with online teaching are improving. Compared with online learning at home during the outbreak period of COVID-19, the campus environment is more friendly and conducive to learning [52]. In addition, compared with the learning burnout of students before the outbreak of COVID-19, the clinical probation and internship of nursing students in this study were terminated. Therefore, the time spent on the clinical practice task was lower than that before the COVID-19 pandemic, and the learning pressure was relatively less than that of online teaching and offline teaching. These reasons may explain the difference in the learning burnout scores of nursing students before and after the COVID-19 pandemic.

Firstly, our correlation analysis shows that, under the normalization of the COVID-19 epidemic, students who are not sophomores, have people in the medical profession at home, have excellent academic performance and prefer nursing are not prone to learning burnout. Some of the participants were freshmen during the study period. Therefore, the senior grades, the academic pressure and employment pressure of freshmen in China were relatively small, while the number of courses for sophomores increased significantly. Coupled with the challenge of participating in the clinical practice process for the first time, there is inadequate preparation to cope with knowledge and skill demands, so the overall academic pressure of sophomore nursing students is higher and the incidence of learning burnout is higher [53,54]. Compared with sophomores, the number of courses and academic pressure for juniors is reduced in China and their adaptability to clinical practice is enhanced. In addition, during the survey, the clinical practice tasks of senior students are nearing the end, and most of the graduates’ work has been implemented, so the academic pressure and employment pressure are relatively small. In summary, due to the gradual and in-depth characteristics of the nursing education process, the sophomores are comparatively faced with the dual challenge of the accumulation of theoretical knowledge and the cultivation of clinical practice ability, which can easily lead to learning burnout. At the same time, the better the academic performance, the higher the recognition of the nursing profession, and the students can invest more interest in learning nursing-related theoretical knowledge and practical skills and achieve a rich harvest. In addition, studies have shown that, compared with non-medical professionals, medical-related professionals have stronger adaptability to COVID-19, so they will be calmer in the face of the threat of COVID-19 and the degree of anxiety and depression is lighter [55,56]. Students with family members in the medical profession are more receptive to the nursing profession, more diligent in their studies and more adaptable and resilient to COVID-19, thereby reducing the impact of COVID-19 on their learning behavior.

Secondly, learning burnout was positively correlated with anxiety and depression (*p* < 0.01) and negatively correlated with academic self-efficacy (*p* < 0.01). Previous studies have shown that mental health problems such as anxiety and depression can damage students’ psychological state of learning, resulting in learning burnout and affecting academic performance [57,58]. According to the theory of self-efficacy, self-efficacy is a protective factor against stress. Specific self-efficacy can affect individual attitudes and behaviors in specific situations, thus affecting their efforts and persistence [38]. The firm belief in the ability to complete learning goals or tasks is a manifestation of a higher level of students’ academic self-efficacy. A meta-analysis found that self-efficacy is a protective factor that reduces burnout among students, teachers, healthcare workers and other service workers [59]. Academic self-efficacy may play an important role in nursing students’ academic burnout. Individuals with low academic self-efficacy are more likely to be affected by learning burnout [37,48].

Finally, the results of this study showed that when academic self-efficacy was included in the regression equation, the direct predictive effect of anxiety and depression on learning burnout became negligible. We analyzed and verified the hypothesis that, the higher the level of anxiety and depression, the more obvious the learning burnout of nursing students; the stronger the academic self-efficacy, the less learning burnout; the academic self-efficacy of nursing students played a mediating role in the relationship between anxiety, depression and academic burnout. All nursing students are inevitably affected by the COVID-19 pandemic, which may damage learning attitudes and behaviors, increase stress, anxiety and depression, and produce panic, helplessness and other negative emotions [31,32,60]. Academic self-efficacy can help students cope with the negative effects of anxiety and depression, thus alleviating academic burnout. The mediating role of academic self-efficacy provides new ideas for reducing nursing students’ academic burnout levels. Compared with the research during the COVID-19 pandemic, under the normalized management of the COVID-19 epidemic the level of learning burnout of nursing students has decreased, but it has a wide range of influence and the score is still at the upper-middle level, especially the problem of learning burnout caused by the low mood of nursing students. This may be due to strict campus epidemic prevention management measures, restrictions on unnecessary access, reduced opportunities to participate in recreational activities, lack of interaction and emotional support and heavy pressure on learning and clinical practice. Therefore, it is necessary to further strengthen the investigation of the emotional status of nursing students under the normalization of the COVID-19 epidemic, to detect and eliminate the learning burnout caused by emotional problems in time [60,61,62]. This requires educators to arrange learning tasks reasonably, do a good job of teaching feedback and communication between teachers and students and find out the problems existing in online learning and the needs of students over time. For the relevant departments of the school, it is necessary to set up a special psychological counselling room, arrange professional psychological experts to provide timely and effective care for students with mental health and regularly carry out psychological lectures.

In summary, given the role of academic self-efficacy in alleviating students’ anxiety and depression symptoms and reducing the incidence of learning burnout, educators should think about how to improve students’ academic self-efficacy based on Bandura’s self-efficacy theory [38]. The study found that the strength of students’ self-efficacy beliefs may more accurately predict their motivation and future academic choices. In the actual teaching process, self-efficacy beliefs can be transformed into students’ self-confidence [63]. It is worth noting that the enhancement of self-confidence not only needs oral praise but also requires the provision of practical opportunities to express their academic achievements and make an objective evaluation. The purpose of the evaluation is to help students summarize the experience of success or failure and to improve students’ self-efficacy beliefs [63]. For students, it is particularly important to evaluate their academic self-efficacy objectively in order to avoid learning burnout. Overestimating or underestimating their academic self-efficacy will affect their learning ability and learning behavior. To sum up, the holistic nursing education system still needs to be reformed to address the lasting impact that COVID-19 may have on the care industry.

## 5. Limitations

In addition, this study also has some limitations. First, the sample selection of this study is limited. The study only investigated the school of nursing of a university in Jiangsu Province, China, and may not represent the actual situation of all nursing students. Secondly, this study was carried out through the ‘Questionnaire Star’ network platform in terms of publishing and recovering the questionnaire, so we are unable to know about the process of filling out the questionnaire and whether the students completed the questionnaire independently, which may affect the quality of the recovered data. Finally, due to the study’s limitations, the impact of learning burnout and the relationship between the various factors need further study and research. It is worth noting that a more comprehensive study of the learning psychology of nursing undergraduates is needed. More multi-center and large-scale surveys are needed to improve the representativeness of the samples and help promote the results to the target population.

## 6. Conclusions

The study found that, under the normalization of the COVID-19 pandemic, the higher the anxiety and depression levels of nursing undergraduates, the stronger their learning burnout and the lower their academic self-efficacy. In addition, academic self-efficacy plays a significant mediating role in the impact of mental health (anxiety and depression) on learning burnout. It has significance for nursing educators that, in today’s coexistence with COVID-19, the learning attitude and learning behavior and mental health status of nursing undergraduates still need attention, especially in developing countries and poor countries. Factors such as the huge pressure of coping with the COVID-19 epidemic, the lack of awareness of psychological care and the shortage of educational resources have caused some countries and regions to have no time to take the psychological problems of their students into account and to neglect those groups of college students who are relatively vulnerable to the impact of COVID-19, especially with regard to the learning burnout and mental health status of nursing students.

## Figures and Tables

**Figure 1 ijerph-20-04194-f001:**
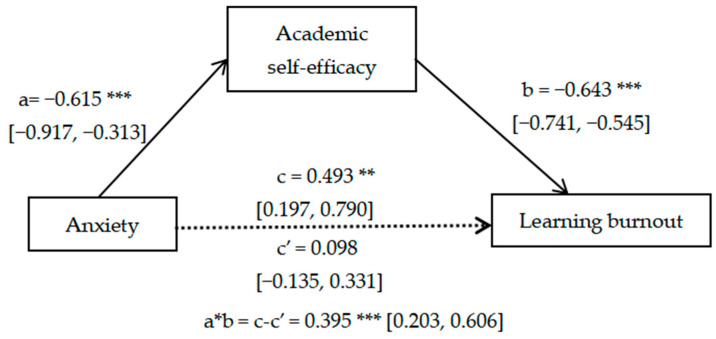
Mediating effect of academic self-efficacy on anxiety and learning burnout. ** *p*-value < 0.01, *** *p*-value < 0.001.

**Figure 2 ijerph-20-04194-f002:**
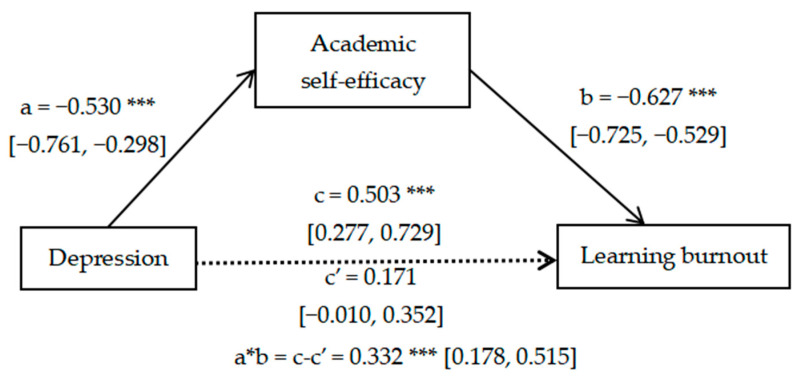
Mediating effect of academic self-efficacy on depression and learning burnout. *** *p*-value < 0.001.

**Table 1 ijerph-20-04194-t001:** Distribution of the learning burnout according to demographic characteristic groups (*n* = 227).

Variables	*n* (%)	Mean ± SD	t/F	*p*
**Gender**				
Male	35 (15.42%)	54.8 ± 1.42	1.155	0.284
Female	192 (84.58%)	53.97 ± 0.732		
**Age**				
<20	46 (20.26%)	52.13 ± 1.729	2.667	0.072
20–22	151 (66.52%)	55.17 ± 0.761		
>22	30 (13.22%)	51.77 ± 1.607		
**Grade**				
Freshman	54 (23.79%)	51.02 ± 1.549	3.029	0.03
Sophomore	51 (22.47%)	56.02 ± 1.379		
Junior	58 (25.56%)	55.72 ± 1.196		
Senior	64 (31.18%)	53.7 ± 1.086		
**Place of Residence**				
urban	72 (31.71%)	51.96 ± 1.161	2.584	0.78
town	66 (29.07%)	54.76 ± 1.357		
rural	89 (39.22%)	55.35 ± 0.93		
**Only child**				
Yes	113 (49.78%)	54.04 ± 0.939	0.322	0.571
No	114 (50.22%)	54.16 ± 0.919		
**Medical professionals** **among relatives**				
Yes	73 (32.16%)	51.66 ± 1.224	6.75	0.01
No	154 (67.84%)	55.26 ± 0.758		
**Academic record**				
outstanding	24 (10.52%)	51.25 ± 1.684	4.526	0.004
good	107 (47.14%)	52.46 ± 0.973		
moderate	87 (38.33%)	56.23 ± 1.028		
poor	9 (4.01%)	60.67 ± 2.867		
**Household income monthly**				
<1000 CNY	9 (4.01%)	56.89 ± 2.659	1.097	0.351
1000–2999 CNY	55 (24.23%)	54.71 ± 1.34		
3000–4999 CNY	83 (36.56%)	54.86 ± 1.086		
>5000 CNY	80 (35.2%)	52.59 ± 1.116		
**Professional Preference**				
love	20 (8.81%)	50.75 ± 2.761	4.246	0.006
prefer	81 (35.68%)	52.19 ± 1.119		
moderate	117 (51.5%)	55.45 ± 0.803		
dislike	9 (4.01%)	61.22 ± 3.915		

**Table 2 ijerph-20-04194-t002:** Means, standard deviations, and correlations of all the measures (*n* = 227).

Variables	Test Score (Mean ± SD)	1	2	3	4	5	6	7	8	9
1.GAD-7	4.60 ± 0.283	-								
2.PHQ-9	5.30 ± 0.366	0.778 **	-							
3. Learning burnout	54.10 ± 0.656	0.213 **	0.281 **	-						
4. Low Mood	21.07 ± 0.336	0.197 **	0.275 **	0.888 **	-					
5. Improper Behavior	16.57 ± 0.237	0.111	0.184 **	0.850 **	0.658 **	-				
6. Low Achievement	16.46 ± 0.22	0.216 **	0.219 **	0.710 **	0.412 **	0.454 **	-			
7. Academic self-efficacy	74.41 ± 0.674	−0.258 **	−0.287 **	−0.672 **	−0.530 **	−0.555 **	−0.599 **	-		
8. Self-efficacy of learning ability	37.72 ± 0.419	−0.241 **	−0.240 **	−0.625 **	−0.478 **	−0.516 **	−0.578 **	0.952 **	-	
9. Self-efficacy of learning behavior	36.69 ± 0.304	−0.241 **	−0.308 **	−0.631 **	−0.517 **	−0.520 **	−0.532 **	0.907 **	0.735 **	-

** *p*-value < 0.01.

**Table 3 ijerph-20-04194-t003:** The Mediation Effect of Academic Self-efficacy in Anxiety, Depression and Learning Burnout (*n* = 227).

Dependent Variable	Independent Variable	R^2^	F	β	SE	t	*p*	95%CI
Academic self-efficacy	Anxiety	0.067	16.092 ^***^	−0.615	0.153	−4.011	<0.001	−0.917–−0.313
Learning burnout	Anxiety	0.454	93.024 ^***^	0.098	0.118	0.829	>0.05	−0.135–0.331
	Academic self-efficacy			−0.643	0.050	−12.938	<0.001	−0.741–−0.545
Total				0.493	0.151	3.275	<0.01	0.197–0.790
Direct				0.098	0.118	0.830	>0.05	−0.135–0.331
Indirect				0.395	0.104	-	-	0.203–0.606
Academic self-efficacy	Depression	0.083	20.253 ^***^	−0.530	0.118	−4.500	<0.001	−0.761–−0.298
Learning burnout	Depression	0.460	95.542 ^***^	0.171	0.092	1.858	>0.05	−0.010–0.352
	Academic self-efficacy			−0.627	0.049	−12.586	<0.001	−0.725–−0.529
Total				0.503	0.115	4.384	<0.001	0.277–0.729
Direct				0.171	0.092	1.858	>0.05	−0.010–0.352
Indirect				0.332	0.085	-	-	0.178–0.515

*** *p*-value < 0.001.

## Data Availability

The data presented in this study are available on request from the corresponding author.

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
