# Peer review of "Relationship between Anxiety, Depression and Learning Burnout of Nursing Undergraduates after the COVID-19 Epidemic: The Mediating Role of Academic Self-Efficacy"

_ijerph, 2023, doi:10.3390/ijerph20054194_

Round 1

Reviewer 1 Report

This is an interesting paper with valuable insight into a much-needed area of research focusing on some mental health indicators of undergraduate nursing students after the COVID-19 epidemic. This is a relatively new research topic since investigations have been focusing rather on the short-term impact of the epidemic instead of its long-term effect like this research. Therefore, this study addresses an existing gap in the pandemic-related studies.

Abstract 
The abstract introduces the content of the paper appropriately. However, it should be in an unstructured form.

Introduction 
The introduced theories and previous research results are in line with the content and direction of the paper.
Line 32: Seemingly, some content is missing ("COVID-19 epidemic. However..."). Please reconsider the very beginning of the introduction as it is not appropriate in this way.
Line 72: instead of "fear, anxiety, and depression of COVID-19", it's rather the "fear, anxiety, and depression  caused by COVID-19"
The aims and hypotheses stated are appropriate and fit the topic and the content of the Introduction well.

Methods
In the Participants section, the authors should refer to the general characteristics of the sample as they introduce it later in the result section (see 3.1. General characteristics)
The introduction of the applied questionnaires is quite detailed.
To improve the Methods section, the applied statistical methods should also be introduced in a separate subchapter, as they are not introduced anywhere in the paper.

Results
Results are introduced clearly. The division of the chapters is logical, and the introduction of the results of the analyses is correct. The tables and figures help the reader to understand the results.

Discussion
The discussion is correctly defined. The authors reflected well on previous findings and theories. This makes the paper coherent.

Conclusions
This section is proper as well. It is very impressive how the authors emphasise the psychological aspects of learning burnout, its manifestation and how universities and academics should act against it. Conclusions are consistent with the evidence and arguments presented previously. However, it would be necessary to reflect on all three hypotheses (detailed in the Introduction).

Limitations
The limitations mentioned are correct. It should be emphasised that results cannot be generalised to the target population while the sample is not representative.

Others
The reference style is not following the requirement of the journal. Please use numbering instead of APA style (https://www.mdpi.com/journal/ijerph/instructions) Tables and figures are informative.

Also, a grammar and stylistic check is necessary (e.g. "compared with" should be "compared to")

Overall, this paper is a valuable study that is worth publishing after minor modifications.

Reviewer 2 Report

It would be convenient to specify some other exploratory validation parameter of the scales, although a confirmatory analysis of the adaptation of the scales to that sample and to that context would give the study more robustness.

Regarding the conclusions, they should be consistent with the objectives of the study and provide answers to the hypotheses.

Some expressions of the conclusions could be reflected as proposals.

Reviewer 3 Report

need consistency in discussing research topics, research design please explain in detail, there are several sentences in the abstract that are difficult to understand, results need to be added to previous research and opinions from the author
